# Characterisation of a Cold Atmospheric Pressure Plasma Torch for Medical Applications: Demonstration of Device Safety

**Adam Bennett [1,2], Takuya Urayama [3], Konstantinos Papangelis [3], Peter Yuen [4] and Nan Yu [5,*]**

1 Surface Engineering and Precision Institute, Cranfield University, Cranfield MK43 0AL, UK; a.d.bennett@cranfield.ac.uk
2 Cranfield Plasma Solutions Ltd., Burton Latimer NN15 5PX, UK
3 Adtec Plasma Technology Co., Ltd., Fukuyama, Hiroshima 712-0942, Japan; t-urayama@adtec-rf.co.jp (T.U.); konstantinos@adtecplasma.com (K.P.)
4 Electro-Optics and Remote Sensing Group, CEWIC, Cranfield University, Shrivenham SN6 8LA, UK; p.yuen@cranfield.ac.uk
5 Institute of Materials and Processes, The University of Edinburgh, Edinburgh EH9 3FB, UK
* Correspondence: nan.yu@ed.ac.uk





**Featured Application: The Adtec SteriPlas is a CE-approved gas plasma medical device with proven anti-bacterial efficacy. The SteriPlas has been used to treat complex and chronic conditions by accelerating healing whilst giving no side effects to the patients. The cold gas plasma technology has been demonstrated to manage infections without the need for antibiotics. Problematic wounds treated include diabetic foot ulcers, which are considered one of the worst wound types due to infection with biofilms. The plasma technology has also been deployed in the field of dermatology to target actinic keratoses, which is a challenging skin condition where conventional therapies often bear undesirable side effects; accelerated healing has again been shown with no side effects.**

**Abstract:** The safety and effectiveness of plasma devices are of crucial importance for medical applications. This study presents the novel design of an atmospheric plasma torch (SteriPlas) and its characterisation. The SteriPlas was characterised to ascertain whether it is safe for application on human skin. The emission spectrum discharged from the SteriPlas was shown to be the same as the emission from the MicroPlaSter Beta. The UV emitted from the SteriPlas was measured, and the effective irradiance was calculated. The effective irradiance enabled the determination of the maximum UV exposure limits, which were shown to be over two hours: significantly longer than the current two-minute treatment time. The use of an extraction system with a higher flow rate appears to reduce slightly the effective irradiance at the treatment area. The NOx and ozone emissions were recorded for both SteriPlas configurations. The NOx levels were shown to be orders of magnitude lower than their safety limits. The ozone emissions were shown to be safe 25 mm from the SteriPlas cage. A discussion of how safety standards differ from one regulatory body to another is given.

**Keywords:** cold atmospheric plasma; biological and physical characterisation; plasma medicine; ultraviolet (UV); reactive oxygen nitrogen species (RONS); wound management; anti-bacterial; plasma healthcare; safety standards

## 1. Introduction

Plasma, discharged under vacuum conditions, underpins the development of the silicon wafer industry, which makes the devices that you are using to read this paper. Atmospheric plasma technology—discharging plasma in the air—has been under development for several decades and has many benefits over vacuum technologies: in surface engineering applications, there is no limit on component size, and processing can be

undertaken much faster as there is no requirement for a vacuum chamber [1]; surface engineering applications include nanometre etching of surfaces [2–4], deposition of nanometre coatings [5] and surface energy modification to improve the bonding of components [6].

Atmospheric plasma generation has many other applications beyond surface engineering, which includes increasing the range planes can fly on a certain amount of fuel [7], increasing the energy efficiency of solar energy plants [8], nanomaterial processing [9], as surgical cutting tools [10], for cleaning of contaminated water supplies [11], and decontamination of microorganisms in medical applications [12].

Clearly, many applications using plasma technology at atmospheric pressure now exist, and more are being invented every year. Considering that these technologies are being used in the open air and medical applications are being applied to humans, it is of paramount importance that these technologies are characterised and shown to be safe. Moreover, considering that these technologies will be deployed around the world, international safety standards must also be defined. Mann et al. in 2016 published a DIN specification (German Standard) for characterising a mini plasma torch; however, the specification is only relevant in Europe; moreover, the method used to characterise the ultraviolet (UV) and ozone lacks transparency [13].

The following work in this study shows the novel design of the plasma torch, an Adtec Healthcare SteriPlas, which is currently the largest CE-approved plasma device for human wound management, and its characterisation. The aim of this work is to show the key characterisation techniques that must be undertaken to demonstrate the safety and effectiveness of the medical plasma device. Furthermore, the UV and reactive oxygen nitrogen species (RONS) safety limits will be discussed in a global context, comparing how standards differ between different regions around the world. Finally, limits will be proposed for an international standard.

## 2. Materials and Methods

### 2.1. Plasma Device

The SteriPlas plasma torch (Adtec Europe Ltd., London, UK) is a bespoke design that consists of six stainless steel electrodes placed inside an aluminium cylinder, which is 135 mm long. The surfaces of the six electrodes are serrated and are equally distributed at a distance of 6 mm from the inner surface of the plasma torch. The diameter of the electrodes and the distance between the electrodes and the surface of the plasma torch are 4 mm. Electrode material and surface topography are extremely important with any plasma torch design. The serrated structure of the SteriPlas electrodes increase the local electric field strength, which aids plasma generation and discharge stability. The tips of the electrodes are situated at 20 mm from the opening of the plasma torch. This patented design enables a relatively large (35 mm diameter) treatment area, which is an order of magnitude larger than other devices currently on the market [14,15]. The unique design of the SteriPlas plasma torch is shown in Figure 1.

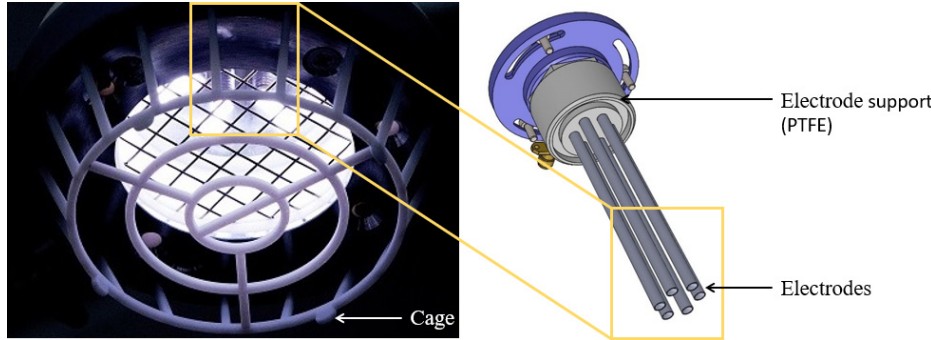

**Figure 1.** Photograph (**left**) of the SteriPlas plasma torch and CAD model (**right**) of the electrode.

## 2.2. Measurements of Plasma Emissions

### 2.2.1. Optical Emission Spectroscopy (OES)

The SteriPlas was ignited at 150 W forward power, at a frequency of 2.45 GHz, with an argon gas flow rate of 2.2 L/min, where the argon gas purity was 99.998%. Then the power was reduced to 80 W forward power, which resulted in circa 2 W reflected power. Cooling air flow to the plasma torch was delivered by either a 55 L/min or 80 L/min air pump, respectively. A fibre optic spectrometer (Ocean Optics USB2000+, Rochester, NY, USA) was connected to a UV-VIS high OH multi-mode optical fibre, which was in turn connected to a UV/Visible Cosine Corrector (Avantes, Apeldoorn, The Netherlands) with a 3.9 mm active area. Data were extracted from the spectrometer by a USB cable to a computer running Ocean View software.

The Ocean Optics USB2000+ fibre optic spectrometer used a 2048-pixel linear charge-coupled device (CCD) array (Toshiba Electronics Asia Pte Ltd., Singapore), with a 5 μm wide 'slit', which was the smallest slit size; hence giving the maximum spectral resolution. The spectral resolution was calculated by taking the spectral range and dividing it by the number of pixels in the CCD array, which provided a relatively high spectral resolution of 161 pm.

Calibration of the spectrometer was performed using a DH-2000 Deuterium Halogen Light Source from Ocean Optics. The deuterium light source was used to calibrate the spectrometer for absolute spectral irradiance measurements. This type of calibration compares the measured spectrum from a calibration light source against the known spectrum from a National Metrology Institute. Such a calibration removes errors in the measurement system due to attenuation of the optical fibre, the potential for solarisation of the optical fibre, noise due to dark current, and noise due to thermal electron effects in the CCD. The calibration process achieves this by comparing the signal being measured to that of a defined standard for a deuterium light source. The calibration process ensures that the absolute irradiance measurements are traceable to National Metrology Institute (NMI) standards.

Next, the absolute spectral irradiance measurements were recorded at varying distances from the treatment area. The treatment area is defined as the cage under the SteriPlas plasma torch. All measurements were made down the centre of the axis of the torch. The 0 mm distance position was at the cage. The experiments were conducted on the SteriPlas with both the 55 L/min and 80 L/min extraction systems sequentially. Figure 2 shows the experimental method.

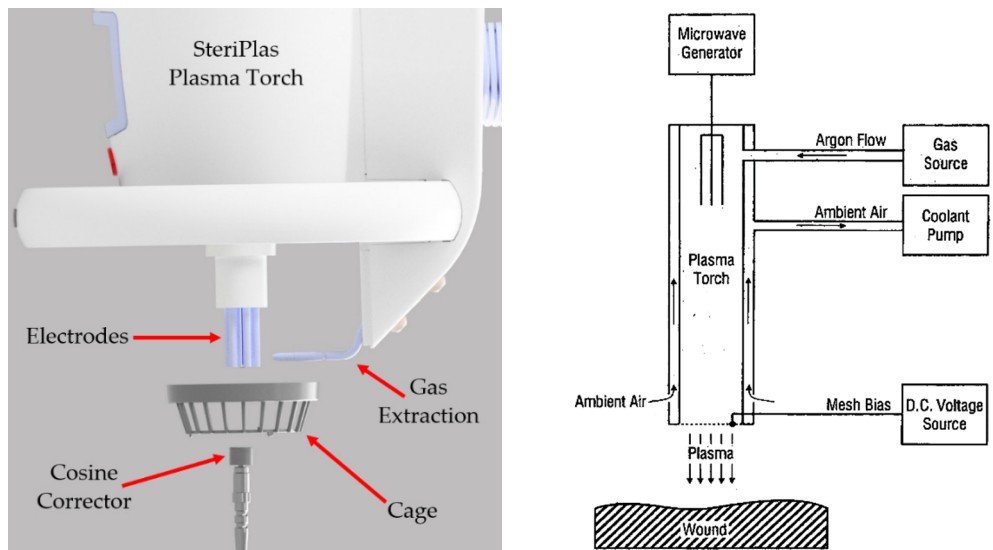

**Figure 2.** Experimental setup of the spectral irradiance measurement (**left**) and the schematic of the plasma torch (**right**) [15].

### 2.2.2. UV Effective Irradiance

The effective irradiance was then calculated using Equation (1). The equation is a simplified version of the integral given in the work of [13], where each absolute spectral irradiance measurement is multiplied by the relative spectral effectiveness for the respective wavelength. This equation also factors in the measurement interval of the spectrometer.

$$E_{eff} = \Sigma E_\lambda \cdot S(\lambda) \cdot \Delta\lambda \tag{1}$$

where $E_{eff}$ is the effective irradiance in $\mu W/cm^2$, $E_\lambda$ is the absolute spectral irradiance from measurements in $\mu W/cm^2/nm$, $S(\lambda)$ is the relative spectral effectiveness, and $\Delta\lambda$ is the bandwidth in nanometres of the measurement intervals in the spectrometer.

Values of $S(\lambda)$ for wavelengths in the 210–400 nm range are calculated using Equations (2)–(4) [16]:

210 nm $\leq \lambda \leq$ 270 nm $\Rightarrow$

$$S(\lambda) = 0.959^{(270-\lambda)} \tag{2}$$

270 nm $< \lambda \leq$ 300 nm $\Rightarrow$

$$S(\lambda) = 1 - 0.36\left(\frac{\lambda - 270}{20}\right)^{1.64} \tag{3}$$

300 nm $< \lambda \leq$ 400 nm $\Rightarrow$

$$S(\lambda) = 0.3 \times 0.736^{(\lambda-300)} + 10^{(2-0.0163\lambda)} \tag{4}$$

The relative spectral effectiveness for wavelengths in the 180–400 nm range is given in the work of [17].

### 2.2.3. NOx and Ozone Measurements

NOx measurements were made using an Enviro Technology 200E Chemiluminescence NOx Analyser, and ozone measurements were made using an Enviro Technology 400E Photometric $O_3$ Analyser. The NOx and ozone measurement systems were set up next to each other, but due to the nature of the gas measurement process, the measurement experiments were run sequentially: first, the NOx data were recorded, and then the ozone data were recorded. Both measurement systems had pipes that extended to the desired measurement locations; however, open pipe ends were not appropriate for this type of experiment as they would not record the maximum gas levels, so a bespoke funnel was 3D printed from PLA+ (a polymer material made from polylactic acid) and attached to the end of the measurement pipe being used. The funnel was designed so that the large opening was circular and matched the diameter of the treatment area (bottom of the cage) of the SteriPlas, which is defined in these experiments as distance = 0 mm.

The NOx and ozone measurement systems both required a 24 h period of calibration with ambient conditions before recording measurements, so both systems were left to run for a 48 h period over the weekend to ensure a full calibration cycle before the SteriPlas was switched on and measurements were recorded. The NOx and ozone measurements were then recorded at the same distances that were used in the UV measurements work package. Five repeats of each experiment were conducted to ensure that the results were consistently repeatable.

## 3. Results

### 3.1. Absolute Spectral Irradiance Measurements

Figure 3 shows an example of the absolute spectral irradiance measured. UV irradiance was present within the UV-A (400 nm to 320 nm) and UV-B (320 nm to 280 nm) ranges but not within the UV-C (280 nm to 200 nm) range. However, the spectrometer was not capable of measuring absolute irradiance below 200 nm wavelengths.

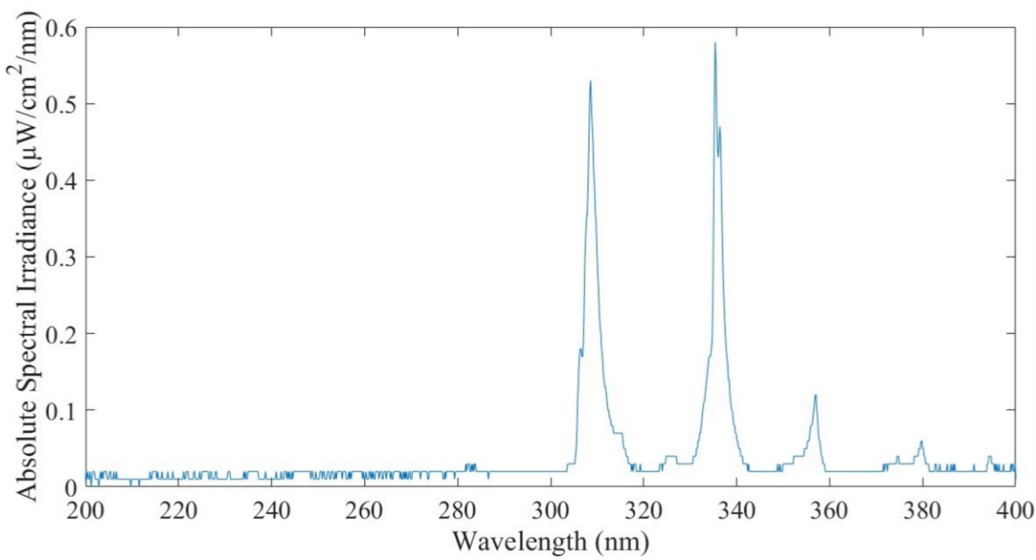

**Figure 3.** Absolute spectral irradiance measurement.

### 3.2. UV Effective Irradiance

Figure 4 shows the effective irradiance emitted from the SteriPlas when either operating with the 55 or 80 L/min extraction system, respectively. The temporal variance of the absolute spectral irradiance measurements was only 1%, and consequently, the error bars in the effective irradiance are too small to quantitatively be shown in the figure. The positioning accuracy of the cosine corrector was 1 mm.

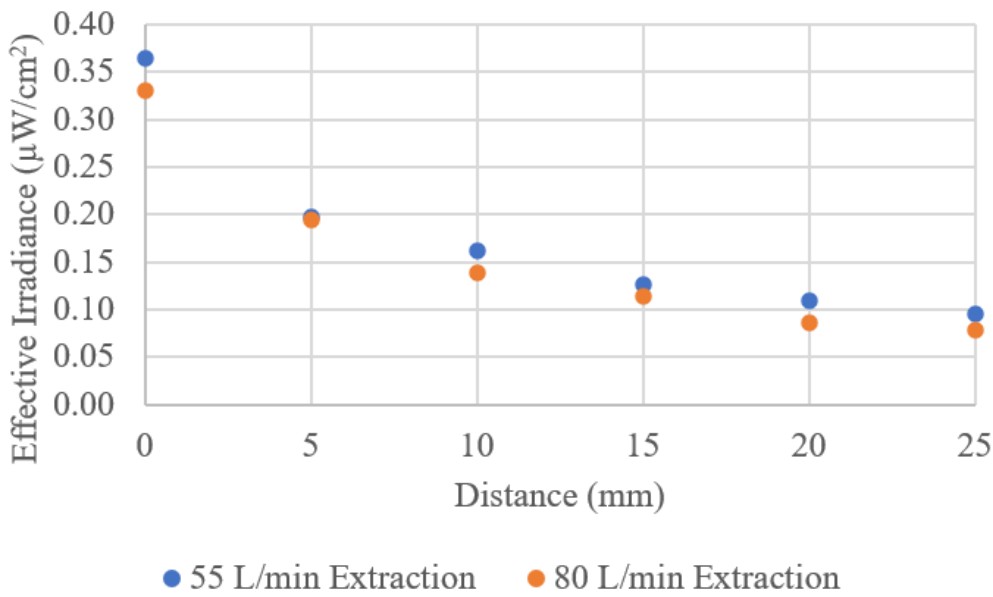

**Figure 4.** Effective irradiance against vertical distance from the SteriPlas torch.

The effective irradiance, in both extraction system configurations, decreases with distance in accordance with theory, which predicts that photonic energy should follow as an inverse relationship with respect to the distance from the source (inverse square for a spherical light source). It should be noted that the 55 L/min extraction system exhibits a slightly higher effective irradiance at the cage (distance = 0 mm) and that this can be explained by a lower rate of extraction; however, it should also be noted that these experiments were not conducted using a precision motion stage and therefore there is a relatively small error in the positioning of the cosine corrector, which is important to

highlight as typical characterisation of medical devices are conducted in healthcare settings and not scientific laboratories.

### 3.3. NOx and Ozone Emission

Figures 5 and 6 show the NOx emitted from the SteriPlas configurations with the 55 and 80 L/min air pump extraction systems, respectively.

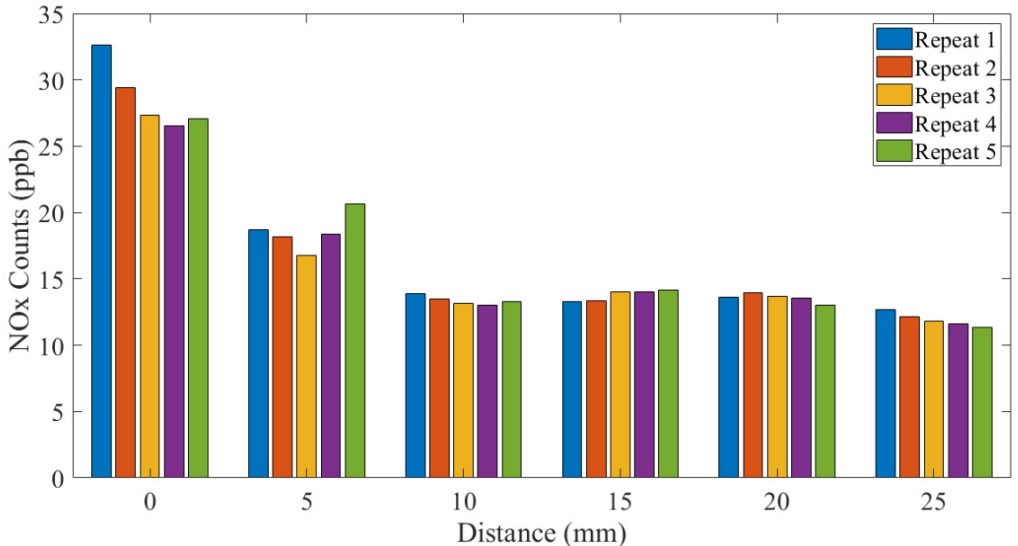

**Figure 5.** NOx measurements: 55 L/min extraction system.

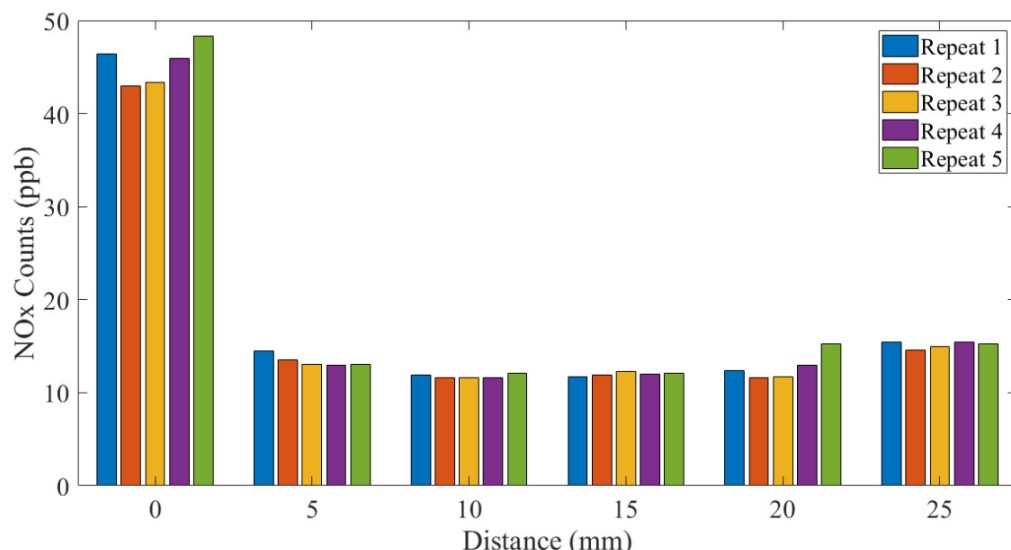

**Figure 6.** NOx measurements: 80 L/min extraction system.

Figures 7 and 8 show a detailed view of the ozone emitted from the SteriPlas configurations with the 55 and 80 L/min air pump systems, respectively. The exposure limits were added to these graphs to highlight that the emissions at 25 mm distance from the SteriPlas cage were safe.

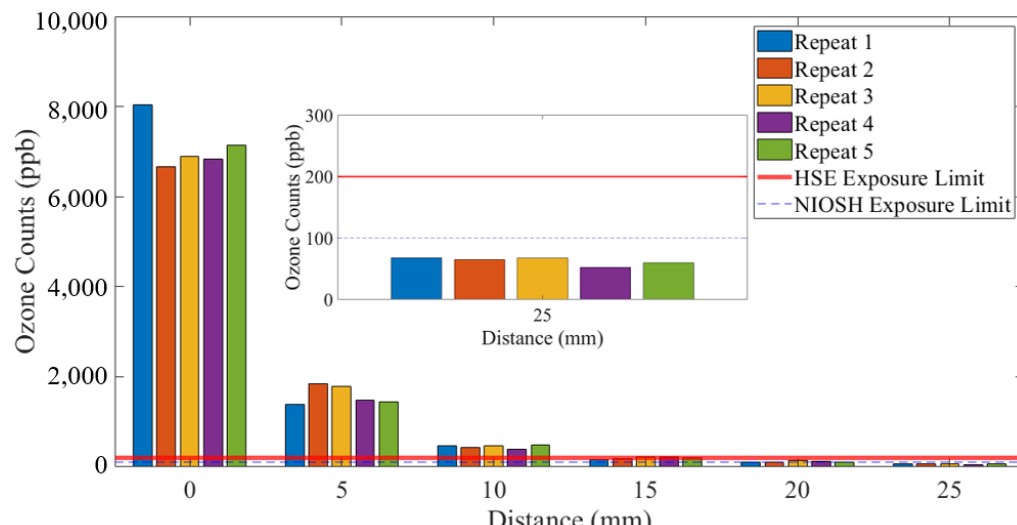

**Figure 7.** Ozone measurements: 55 L/min extraction system.

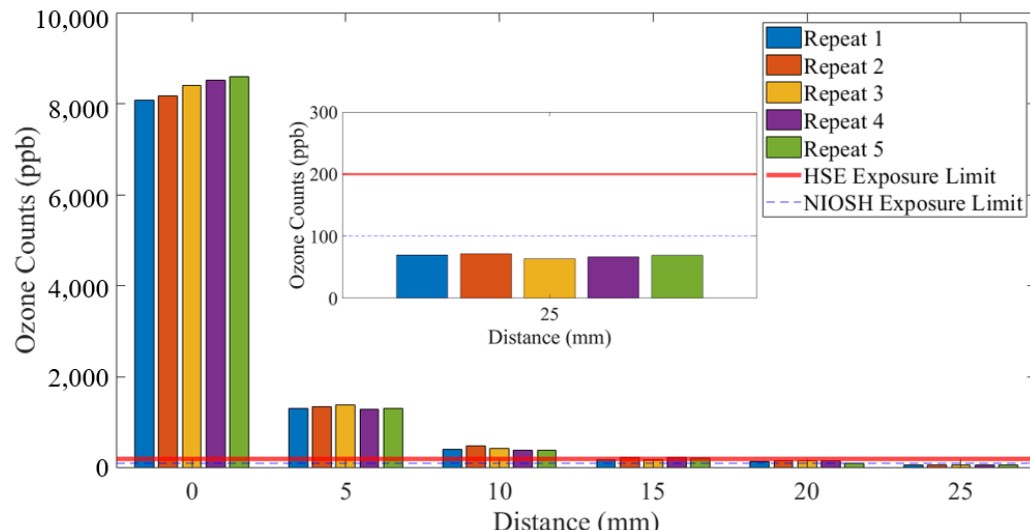

**Figure 8.** Ozone measurements: 80 L/min extraction system.

## 4. Discussion

### 4.1. UV Measurements

The effective irradiance at the cage (Distance = 0 mm), which is the treatment area in contact with the patient, was 0.36 $\mu$W/cm$^2$ for the SteriPlas with the 55 L/min extraction system and 0.33 $\mu$W/cm$^2$ for the SteriPlas with the 80 L/min extraction system. These values show that the air pump with a higher extraction flow rate lowers the UV energy incident upon the patient. Furthermore, the effective irradiance was calculated to provide a more accurate value for the maximum safe exposure limits; previously, Adtec had simply added together the UV power emitted at different wavelengths, which is a safe practice, erring on the side of caution, but does not show the maximum time a human may be exposed to the SteriPlas.

The International Commission on Non-Ionizing Radiation Protection (ICNIRP) states that for the most sensitive, non-pathologic, skin photo-types, ultraviolet radiant exposure in the spectral region of 180 to 400 nm upon the unprotected skin should not exceed 30 J/m$^2$, effective spectrally weighted using the spectral weighting factors calculated from Equations (2)–(4) and shown in the work of [17].

Now, 30 J/m$^2$ = 30 W/m$^2$ = 3000 µW/cm$^2$ if the exposure time is set to 1 s in a given 24 h period. Therefore, by simply dividing 3000 µW/cm$^2$ by the effective irradiances above, the maximum exposure time in a given 24 h period is obtained at the treatment area under the SteriPlas: maximum exposure time (SteriPlas 55 L/min extraction system) = 3000 µW/cm$^2$/0.36 µW/cm$^2$ = 2 h, 18 min, and 53 s; maximum exposure time (SteriPlas 80 L/min extraction system) = 3000 µW/cm$^2$/0.33 µW/cm$^2$ = 2 h, 31 min, and 30 s.

### 4.2. NOx and Ozone Measurements

Exposure limits for a given airborne chemical, such as ozone, vary depending upon the regulator. Different countries have different limits, and these limits change over time as standards are redefined. Moreover, a given geographical location may be subject to more than one regulatory body, which can cause confusion. Internationally respected regulatory bodies include the U.S. National Institute for Occupational Safety and Health (NIOSH), U.K. Health and Safety Executive (HSE), European Parliament and The Council of The European Union, and the World Health Organisation. Previous work conducted by Adtec used the NIOSH and HSE standards.

The NIOSH standards [18] define the nitrogen monoxide (NO) exposure limit as 25 ppm, nitrogen dioxide (NO$_2$) exposure limit as 1 ppm, and ozone (O$_3$) exposure limit as 0.1 ppm. The HSE standards [19] define the nitrogen monoxide (NO) eight-hour exposure limit as 25 ppm for the underground mining and tunnelling industries and 2 ppm for all other activities, there is no 15-min exposure limit; nitrogen dioxide (NO$_2$) eight-hour exposure limit as 0.5 ppm, the 15-min exposure limit is 1 ppm, and ozone (O$_3$) 15-min exposure limit as 0.2 ppm, there is no eight-hour exposure limit.

Clearly, just comparing these two standards, there is inconsistency in limits and the time periods over which those limits exist. The NOx measurements shown in this paper are a sum of the nitrogen monoxide (NO) and nitrogen dioxide (NO$_2$) counts added together and are orders of magnitude lower than the safety limits defined above. The ozone (O$_3$) measurements shown in Figures 7 and 8 show that 25 mm from the SteriPlas cage levels are below the NIOSH and HSE limits; it should be noted that the NOx and ozone limits apply to the inhalation of the gases and not to the exposure to human skin, as long as no patient or operator has their mouth and/or nose within 25 mm of the SteriPlas cage it will be completely safe.

### 4.3. Future Work

Authors understand that it would be nice to have a preliminary results comparison between this equipment and others already in use in the medical field. Reference [13] shows a similar approach taken, but is on different plasma technology and is not explained fully, so it cannot be compared. Other spectrometers and gas sensors exist, but without using them on this plasma torch, no comparison can be made. For example, the authors focused on the characterisations of ICP torches and MIP torches from previous studies [20,21]. This highlights the importance of this research: creating a standard characterisation methodology for plasma technology, which may be undertaken with any spectrometers and gas sensors but must be conducted in a calibrated fashion in accordance with the principles of precision engineering.

### 5. Conclusions

The SteriPlas has been tested to ascertain whether it is still safe for application within medical trials, and the results show that it meets the standards of internationally respected regulatory bodies. The UV emitted from the SteriPlas was measured, and the effective irradiance was calculated. The unit with the 55 L/min extraction system exhibits a slightly higher effective irradiance at the treatment area, and this is hypothesised to be due to the lower gas extraction. The relatively low effective irradiance for both SteriPlas configurations results in a safe plasma treatment time of over two-hours from a UV perspective, and this

is significantly longer than the current two-minute treatment time that is used in clinical settings. The NOx and ozone emissions were recorded for both SteriPlas configurations. The NOx levels were shown to be orders of magnitude lower than their safety limits. The ozone emissions were shown to be safe 25 mm from the SteriPlas cage.

**Author Contributions:** Conceptualisation, methodology, validation, formal analysis, investigation, A.B.; resources, T.U., K.P. and P.Y. data curation, N.Y.; writing—original draft preparation, A.B. and N.Y.; writing—review and editing, T.U., K.P. and P.Y. All authors have read and agreed to the published version of the manuscript.

**Funding:** This research was funded by Adtec Plasma Technology.

**Informed Consent Statement:** Not applicable.

**Acknowledgments:** The authors wish to thank Adtec Plasma Technology, Adtec Europe, Adtec Healthcare, and Cranfield University for the loan of state-of-the-art equipment to undertake the experiments shown in this paper.

**Conflicts of Interest:** The authors declare no conflict of interest.

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
