# Peer review of "Characterisation of a Cold Atmospheric Pressure Plasma Torch for Medical Applications: Demonstration of Device Safety"

_applsci, doi:10.3390/app112411864_

Round 1
Reviewer 1 Report
This paper reports a novel design of an atmospheric plasma torch (SteriPlas) and its characterisation. The SteriPlas was characterised to ascertain whether it is safe for application on human skin. The emission spectrum discharged from the SteriPlas was shown to be the same as the emission from the MicroPlaSter Beta. The UV emitted from the SteriPlas was measured and the effective irradiance was calculated, which enabled the determination of the maximum UV exposure limits. It is an interesting paper, contains novel aspects which provides interesting results with appropriate references, therefore it is recommended for publication after a few minor revisions shown below.
- In the introduction, please explain: DIN-specification for characherising a mini plasma torch
- In figure1, then components of the SteriPlas plasma torch should be identified
- The sentence in line 100-103, is confused; should be reformulated
- Line 143, define PLA+
- Nota bene, appears several times in the manuscript; I suggest to substitute these two words, I think there are not adequate for a scientific paper.
Author Response
Dear Reviewer:
We sincerely appreciate your kind decision and suggestions for improving this paper. The main corrections in the paper and the response to the comments are as following:
Comment 1: In the introduction, please explain: DIN-specification for characterising a mini plasma torch
Response: DIN is a German, National Standard. It does not apply beyond the country. The purpose of this paper is to clearly explain a methodology, which may be copied around the world and may be the basis for an International Standard, as it factors in exposure limits in different regions. Audience could refer [13] to further understand this.
Correction: add note: (German Standard) right after DIN-specification.
Comment 2: In figure1, then components of the SteriPlas plasma torch should be identified
Response: thanks for this valuable suggestion. The figure 1 is re-produced.
Correction: please see the figure in the manuscript. Caption is revised in manuscript.
Comment 3: The sentence in line 100-103, is confused; should be reformulated
Response: thanks for this valuable suggestion. This sentence is revised as below. Original: This type of calibration factors in the optical efficiency of the spectrometer over the given spectral range, the attenuation of the optical fibre, the potential for solarisation of the optical fibre, noise due to dark current, noise due to thermal electron effects in the CCD, and background signals in the room when the plasma is off.
Correction: This type of calibration compares the measured spectrum from a calibration light source against the known spectrum from a National Metrology Institute. Such a calibration removes errors in the measurement system due to attenuation of the optical fibre, the potential for solarisation of the optical fibre, noise due to dark current, and noise due to thermal electron effects in the CCD.
Comment 4: Line 143, define PLA+
Response: A polymer material made from polylactic acid, which is commonly used in 3D printing.
Correction: so a bespoke funnel was 3D printed from PLA+ (A polymer material made from pol-ylactic acid)…
Comment 5: Nota bene, appears several times in the manuscript; I suggest to substitute these two words, I think there are not adequate for a scientific paper.
Response: Valuable suggestion is fully accepted.
Correction: Delete all NB from the manuscript.
Reviewer 2 Report
Dear Authors
The manuscript is well written and easy to read.
Some minor improvements can be done, such as:
- et al. must be in italic once it is Latin in all manuscript;
- Abbreviations don’t have plural, therefore RONS should be RON
- The patent number should be added to the manuscript
- All equipment’s and software must have between brackets (Producer, City, Country), and the software the version and for which operating system.
It would be nice to have a preliminary results comparation between this equipment and others already in use in medical field.
Author Response
We sincerely appreciate your kind decision and suggestions for improving this paper. The main corrections in the paper and the response to the comments are as following:
Comment 1: et al. must be in italic once it is Latin in all manuscript;
Response: Valuable suggestion is fully accepted.
Correction: add note: (German Standard) right after DIN-specification.
Comment 2: Abbreviations don’t have plural, therefore RONS should be RON
Response: Not plural, it stands for Reactive Oxygen Nitrogen Species
Comment 3: The patent number should be added to the manuscript
Response: This patent number is added in reference [15]
Comment 4: All equipment’s and software must have between brackets (Producer, City, Country), and the software the version and for which operating system.
Response: Valuable suggestion is fully accepted.
Correction: Please find them in sections 2.1 and 2.2.
Comment 5: It would be nice to have a preliminary results comparation between this equipment and others already in use in medical field.
Response: Reference in paper shows a similar approach taken, but is on different plasma technology and is not explained fully, so cannot be compared. Other spectrometers and gas sensors exist, but without using them on this plasma torch no comparison can be made. This highlights the importance of this paper: creating a standard characterisation methodology for plasma technology, which may be undertaken with any spectrometer etc. but must be conducted in a precision, calibrated, fashion.